# Meta-Learning through Hebbian Plasticity in Random Networks

**Elias Najarro and Sebastian Risi**
IT University of Copenhagen
2300 Copenhagen, Denmark
enaj@itu.dk, sebr@itu.dk

## Abstract

Lifelong learning and adaptability are two defining aspects of biological agents. Modern reinforcement learning (RL) approaches have shown significant progress in solving complex tasks, however once training is concluded, the found solutions are typically static and incapable of adapting to new information or perturbations. While it is still not completely understood how biological brains learn and adapt so efficiently from experience, it is believed that synaptic plasticity plays a prominent role in this process. Inspired by this biological mechanism, we propose a search method that, instead of optimizing the weight parameters of neural networks directly, only searches for synapse-specific Hebbian learning rules that allow the network to continuously self-organize its weights during the lifetime of the agent. We demonstrate our approach on several reinforcement learning tasks with different sensory modalities and more than 450K trainable plasticity parameters. We find that starting from completely random weights, the discovered Hebbian rules enable an agent to navigate a dynamical 2D-pixel environment; likewise they allow a simulated 3D quadrupedal robot to learn how to walk while adapting to morphological damage not seen during training and in the absence of any explicit reward or error signal in less than 100 timesteps. Code is available at https://github.com/enajx/HebbianMetaLearning.

## 1   Introduction

Agents controlled by neural networks and trained through reinforcement learning (RL) have proven to be capable of solving complex tasks [1–3]. However once trained, the neural network weights of these agents are typically static, thus their behaviour remains mostly inflexible, showing limited adaptability to unseen conditions or information. These solutions, whether found by gradient-based methods or black-box optimization algorithms, are often immutable and overly specific for the problem they have been trained to solve [4, 5]. When applied to a different tasks, these networks need to be retrained, requiring many extra iterations.

Unlike artificial neural networks, biological agents display remarkable levels of adaptive behavior and can learn rapidly [6, 7]. Although the underlying mechanisms are not fully understood, it is well established that synaptic plasticity plays a fundamental role [8, 9]. For example, many animals can quickly walk after being born without any explicit supervision or reward signals, seamlessly adapting to their bodies of origin. Different plasticity-regulating mechanisms have been suggested which can be encompassed in two main ideal-type families: end-to-end mechanisms which involve top-down feedback propagating errors [10] and local mechanisms, which solely rely on local activity in order to regulate the dynamics of the synaptic connections. The earliest proposed version of a purely local mechanism is known as *Hebbian plasticity*, which in its simplest form states that the synaptic strength between neurons changes proportionally to the correlation of activity between them [11].

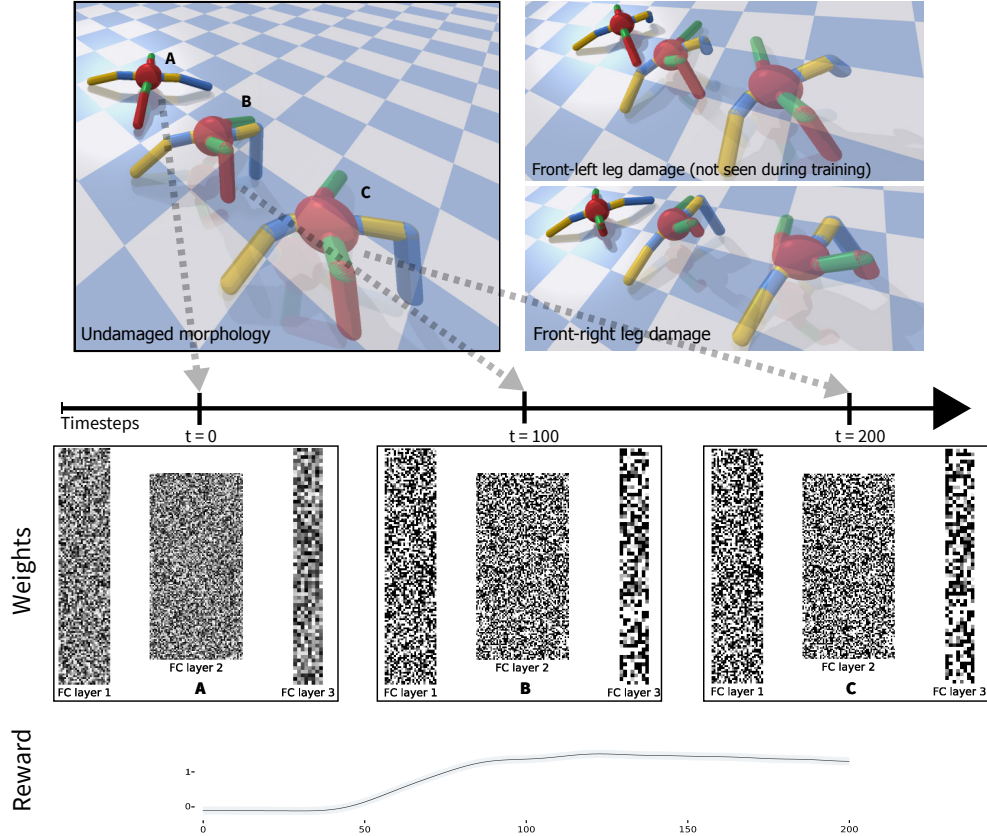

Figure 1: *Hebbian Learning in Random Networks.* Starting from random weights, the discovered learning rules allow fast adaptation to different morphological damage without an explicit reward signal. The figure shows the weights of the network at three different timesteps (A, B, C) during the lifetime of the robot with standard morphology (top-left). Each column represents the weights of each of the network layers at the different timesteps. At t=0 (A) the weights of the network are initialised randomly by sampling from an uniform distribution $\mathbf{w} \in U[-0.1, 0.1]$, thereafter their dynamics are determined by the evolved Hebbian rules and the sensory input from the environment. After a few timesteps the quadruped starts to move which reflects in an increase in the episodic reward (bottom row). The network with the same Hebbian rules is able to adapt to robots with varying morphological damage, even ones not seen during training (top right).

The rigidity of non-plastic networks and their inability to keep learning once trained can partially be attributed to them traditionally having both a fixed neural architecture and a static set of synaptic weights. In this work we are therefore interested in algorithms that search for plasticity mechanisms that allow agents to adapt during their lifetime [12–15]. While recent work in this area has focused on determining both the weights of the network and the plasticity parameters, we are particularly intrigued by the interesting properties of randomly-initialised networks in both machine learning [16–18] and neuroscience [19]. Therefore, we propose to search for plasticity rules that work with randomly initialised networks purely based on a process of self-organisation.

To accomplish this, we optimize for connection-specific Hebbian learning rules that allow an agent to find high-performing weights for non-trivial reinforcement learning tasks without any explicit reward during its lifetime. We demonstrate our approach on two continuous control tasks and show that such a network reaches a higher performance than a fixed-weight network in a vision-based RL task. In a 3-D locomotion task, the Hebbian network is able to adapt to damages in the morphology of a simulated quadrupedal robot which has not been seen during training, while a fixed-weight network fails to do so. In contrast to fixed-weight networks, the weights of the Hebbian networks continuously vary during the lifetime of the agent; the evolved plasticity rules give rise to the emergence of

an attractor in the weight phase-space, which results in the network quickly converging to high-performing dynamical weights.

We hope that our demonstration of random Hebbian networks will inspire more work in neural plasticity that challenges current assumptions in reinforcement learning; instead of agents starting deployment with finely-tuned and frozen weights, we advocate for the use of more dynamical neural networks, which might display dynamics closer to their biological counterparts. Interestingly, we find that the discovered Hebbian networks are remarkably robust and can even recover from having a large part of their weights zeroed out.

In this paper we focus on exploring the potential of Hebbian plasticity to master reinforcement learning problems. Meanwhile, artificial neural networks (ANNs) have been the object of great interest by neuroscientists for being capable of explaining some neurobiological data [20], while at the same time being able to perform certain visual cognitive tasks at a human-level. Likewise, demonstrating how random networks – solely optimised through local rules – are capable of reaching competitive performance in complex tasks may contribute to the pool of plausible models for understanding how learning occurs in the brain. Finally, we hope this line of research will further help promoting ANN-based RL frameworks to study how biological agents learn [21].

## 2   Related work

**Meta-learning**. The aim in meta-learning or learning-to-learn [22, 23] is to create agents that can learn quickly from ongoing experience. A variety of different methods for meta-learning already exist [24–29]. For example, Wang et al. [27] showed that a recurrent LSTM network [30] can learn to reinforcement learn. In their work, the policy network connections stay fixed during the agent's lifetime and learning is achieved through changes in the hidden state of the LSTM. While most approaches, such as the work by Wang et al. [27], take the environment's reward as input in the inner loop of the meta-learning algorithms (either as input to the neural network or to adjust the network's weights), we do not give explicit rewards during the agent's lifetime in the work presented here.

Typically, during meta-training, networks are trained on a number of different tasks and then tested on their ability to learn new tasks. A recent trend in meta-learning is to find good initial weights (e.g. through gradient descent [28] or evolution [29]), from which adaptation can be performed in a few iterations. One such approach is Model-Agnostic Meta-Learning (MAML) [28], which allows simulated robots to quickly adapt to different goal directions. Hybrid approaches bringing together gradient-based learning with an unsupervised Hebbian rules have also proven to improve performance on supervised-learning tasks [31].

A less explored meta-learning approach is the evolution of *plastic* networks that undergo changes at various timescales, such as in their neural connectivity while experiencing sensory feedback. These *evolving plastic networks* are motivated by the promise of discovering principles of neural adaptation, learning, and memory [13]. They enable agents to perform a type of meta-learning by adapting during their lifetime through evolving recurrent networks that can store activation patterns [32] or by evolving forms of local Hebbian learning rules that change the network's weights based on the correlated activation of neurons ("what fires together wires together"). Instead of relying on Hebbian learning rules, early work [14] tried to explore the optimization of the parameters of a parameterised learning rule that is applied to all connections in the network. Most related to our approach is early work by Floreano and Urzelai [33], who explored the idea of starting networks with random weights and then applying Hebbian learning. This approach demonstrated the promise of evolving Hebbian rules but was restricted to only four different types of Hebbian rules and small networks (12 neurons, 144 connections) applied to a simple robot navigation task.

Instead of training local learning rules through evolutionary optimization, recent work showed it is also possible to optimize the plasticity of individual synaptic connections through gradient descent [15]. However, while the trainable parameters in their work only determine how plastic each connection is, the black-box optimization approach employed in this paper allows each connection to implement its own Hebbian learning rule.

**Self-Organization**. Self-organization plays a critical role in many natural systems [34] and is an active area of research in complex systems. It also recently gaining more prominence in machine learning, with graph neural networks being a noteworthy example [35]. The recent work by Mord-

vintsev et al. [36] on growing cellular automata through local rules encoded by a neural network has interesting parallels to the work we present here; in their work the growth of 2D images relies on self-organization while in our work it is the network's weights themselves that self-organize. A benefit of self-organizing systems is that they are very robust and adaptive. The goal in our proposed approach is to take a step towards similar levels of robustness for neural network-based RL agents.

**Neuroscience**. In biological nervous systems, the weakening and strengthening of synapses through synaptic plasticity is assumed to be one of the key mechanisms for long-term learning [8, 9]. Evolution shaped these learning mechanisms over long timescales, allowing efficient learning during our lives. What is clear is that the brain can rewire itself based on experiences we undergo during our lifetime [37]. Additionally, animals are born with a highly structured brain connectivity that allows them to learn quickly form birth [38]. However, the importance of random connectivity in biological brains is less well understood. For example, random connectivity seems to play a critical role in the prefrontal cortex [39], allowing an increase in the dimensionality of neural representations. Interestingly, it was only recently shown that these theoretical models matched experimental data better when random networks were combined with simple Hebbian learning rules [19].

The most well-known form of synaptic plasticity occurring in biological spiking networks is spike-timing-dependent plasticity (STDP). On the other hand, artificial neural networks have continuous outputs which are usually interpreted as an abstraction of spiking networks in which the continuous output of each neuron represents a *spike-rate coding* average –instead of *spike-timing coding*– of a neuron over a long time window or, equivalent, of a subset of spiking neurons over a short time window; in this scenario, the relative timing of the pre and post-synaptic activity does not play a central role anymore [40, 41]. Spike-rate-dependent plasticity (SRDP) is a well documented phenomena in biological brains [42, 43]. We take inspiration from this work, showing that random networks combined with Hebbian learning can also enable more robust meta-learning approaches.

## 3 Meta-learning through Evolved Local Learning Rules

The main steps of our approach can be summarized as follows: (**1**) An initial population of neural networks with random synapse-specific learning rules is created, (**2**) each network is initialised with random weights and evaluated on a task based on its accumulated episodic reward, with the network weights changing at each timestep following the discovered learning rules, and (**3**) a new population is created through an evolution strategy [44], moving the learning-rule parameters towards rules with higher cumulative rewards. The algorithm then starts again at (**2**), with the goal to progressively discover more and more efficient learning rules that can work with arbitrary initialised networks.

In more detail, the synapse-specific learning rules in this paper are inspired by biological Hebbian mechanisms. We use a generalized Hebbian ABCD model [45, 46] to control the synaptic strength between the artificial neurons of relatively simple feedforward networks. Specifically, the weights of the agent are randomly initialized and updated during its lifetime at each timestep following:

$$\Delta w_{ij} = \eta_w \cdot (A_w o_i o_j + B_w o_i + C_w o_j + D_w), \tag{1}$$

where $w_{ij}$ is the weight between neuron $i$ and $j$, $\eta_w$ is the evolved learning rates, evolved correlation terms $A_w$, evolved presynaptic terms $B_w$, evolved postsynaptic terms $C_w$, with $o_i$ and $o_j$ being the presynaptic and postsynaptic activations respectively. While the coefficients $A, B, C$ explicitly determine the local dynamics of the network weights, the evolved coefficient $D$ can be interpreted as an individual inhibitory/excitatory bias of each connection in the network. In contrast to previous work, our approach is not limited to uniform plasticity [47, 48] (i.e. each connection has the same amount of plasticity) or being restricted to only optimizing a connection-specific plasticity value [15]. Instead, building on the ability of recent evolution strategy implementations to scale to a large number of parameters [44], our approach allows each connection in the network to have both a different learning rule and learning rate.

We hypothesize that this Hebbian plasticity mechanism should give rise to the emergence of an attractor in weight phase-space, which leads the randomly-initialised weights of the policy network to quickly converge towards high-performing values, guided by sensory feedback from the environment.

## 3.1 Optimization details

The particular population-based optimization algorithm that we are employing is an evolution strategy (ES) [49, 50]. ES have recently shown to reach competitive performance compared to other deep reinforcement learning approaches across a variety of different tasks [44]. These black-box optimization methods have the benefit of not requiring the backpropagation of gradients and can deal with both sparse and dense rewards. Here, we adapt the ES algorithm by Salimans et al. [44] to not optimize the weights directly but instead finding the set of Hebbian coefficients that will dynamically control the weights of the network during its lifetime based on the input from the environment.

In order to evolve the optimal local learning rules, we randomly initialise both the policy network's weights $\mathbf{w}$ and the Hebbian coefficients $\mathbf{h}$ by sampling from an uniform distribution $\mathbf{w} \in$ U[-0.1, 0.1] and $\mathbf{h} \in$ U[0, 1] respectively. Subsequently we let the ES algorithm evolve $\mathbf{h}$, which in turn determines the updates to the policy network's weights at each timestep through Equation 1.

At each evolutionary step $t$ we compute the task-dependent fitness of the agent $F(\mathbf{h_t})$, we populate a new set of $n$ candidate solutions by sampling normal noise $\epsilon_i = \mathcal{N}(0, 1)$ and adding it to the current best solution $\mathbf{h_t}$, subsequently we update the parameters of the solution based on the fitness evaluation of each of the $i \in n$ candidate solutions:

$$\mathbf{h_{t+1}} = \mathbf{h_t} + \frac{\alpha}{n\sigma} \sum_{i=1}^{n} F_i \cdot (\mathbf{h_t} + \sigma\epsilon_i),$$

where $\alpha$ modulates how much the parameters are updated at each generation and $\sigma$ modulates the amount of noise introduced in the candidate solutions. It is important to note that during its lifetime the agent does not have access to this reward.

We compare our Hebbian approach to a standard fixed-weight approach, using the same ES algorithm to optimise either directly the weights or learning rule parameters respectively. All the code necessary to evolve both the Hebbian networks as well as the static networks with the ES algorithm is available at `https://github.com/enajx/HebbianMetaLearning`.

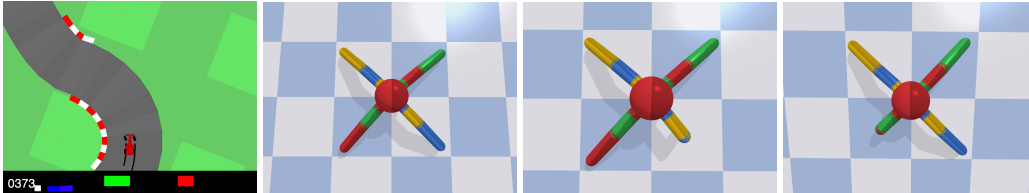

Figure 2: *Test domains.* The random Hebbian network approach introduced in this paper is tested on the `CarRacing-v0` environment [51] and a quadruped locomotion task. In the robot tasks, the same network has to adapt to three morphologies while only seeing two of them during the training phase (standard Ant-v0 morphology, morphology with damaged right front leg and unseen morphology with damaged left front leg) without any explicit reward feedback.

## 4 Experimental Setups

We demonstrate our approach on two continuous control environments with different sensory modalities (Fig. 2). The first is a challenging vision-based RL task, in which the goal is to drive a racing car through procedurally generated tracks as fast possible. While not appearing too complicated, the tasks was only recently solved (achieving a score of more than 900 averaged over 100 random rollouts) [52–54]. The second domain is a complex 3-D locomotion task that controls a four-legged robot [55]. Here the information of the environment is represented as a one-dimensional state vector.

**Vision-based environment** As a vision-based environment, we use the `CarRacing-v0` domain [51], build with the Box2D physics engine. The output state of the environment is resized and normalised, resulting in a observational space of 3 channels (RGB) of 84×84 pixels each. The policy network consists of two convolutional layers, activated by hyperbolic tangent and interposed by pooling layers which feed a 3-layers feedforward network with [128, 64, 3] nodes per layer with no bias. This network has 92,690 weight parameters, 1,362 corresponding to the convolutional layers and

91,328 to the fully connected ones. The three network outputs control three continuous actions (left/right steering, acceleration, break). Under the ABCD mechanism this results in 456,640 Hebbian coefficients including the lifetime learning rates $\eta$.

In this environment, only the weights of the fully connected layers are controlled by the Hebbian plasticity mechanism, while the 1,362 parameters of the convolutional layers remain static during the lifetime of the agent. The reason being that there is no natural definition of what the presynaptic and postsynaptic activity of a convolution filter may be, hence making the interpretation of Hebbian plasticity for convolutional layers challenging. Furthermore, previous research on the human visual cortex indicates that the representation of visual stimuli in the early regions of the ventral stream are compatible with the representations of convolutional layers trained for image recognition [56], therefore suggesting that the variability of the parameters of convolutional layers should be limited. The evolutionary fitness is calculated as -0.1 every frame and +1000/$N$ for every track tile visited, where $N$ is the total number of tiles in the generated track.

**3-D Locomotion Task** For the quadruped, we use a 3-layer feedforward network with $[128, 64, 8]$ nodes per layer, no bias and hyperbolic tangent as activation function. This architectural choice leads to a network with 12,288 synapses. Under the ABCD plastic mechanism, which has 5 coefficients per synapse, this translates to a set of 61,440 Hebbian coefficients including the lifetime learning rates $\eta$. For the state-vector environment we use the open-source Bullet physics engine and its pyBullet python wrapper [57] that includes the "Ant" robot, a quadruped with 13 rigid links, including four legs and a torso, along with 8 actuated joints [58]. It is modeled after the ant robot in the MuJoCo simulator [59] and constitutes a common benchmark in RL [28]. The robot has an input size of 28, comprising the positional and velocity information of the agent and an action space of 8 dimensions, controlling the motion of each of the 8 joints. The fitness function of the quadruped agent selects for distance travelled during a period of 1,000 timesteps along a fixed axis.

The parameters used for the ES algorithm to optimize both the Hebbian and static networks are the following: a population size 200 for the `CarRacing-v0` domain and size 500 for the quadruped, reflecting the higher complexity of this domain. Other parameters were the same for both domains and reflect typical ES settings (ES algorithms are typically more robust to different hyperparameters than other RL approaches [44]), with a learning rate $\alpha$=0.2, $\alpha$ decay=0.995, $\sigma$=0.1, and $\sigma$ decay=0.999. These hyperparameters were found by trial-and-error and worked best in prior experiments.

## 4.1 Results

For each of the two domains, we performed three independent evolutionary runs (with different random seeds) for both the static and Hebbian approach. We performed additional ablation studies on restricted forms of the generalised Hebbian rule, which can be found in the Appendix.

**Vision-based Environment** To test how well the evolved solutions generalize, we compare the cumulative rewards averaged over 100 rollouts for the highest-performing Hebbian-based approach and traditional fixed-weight approach. The set of local learning rules found by the ES algorithm yield a reward of 872$\pm$11, while the static-weights solution only reached a performance of 711$\pm$16. The numbers for the Hebbian network are slightly below the performance of the state-of-the-art approaches in this domain which rely on additional neural attention mechanisms (914$\pm$15 [54]), but on par with deep RL approaches such as PPO (865$\pm$159 [54]). The competitive performance of the Hebbian learning agent is rather surprising, since it starts every one of the 100 rollouts with completely different random weights but through the tuned learning rules it is able to adapt quickly. While the Hebbian network takes slightly longer to reach a high training performance, likely because of the increased parameter space (see Appendix), the benefits are a higher generality when tested on procedurally generated tracks not seen during training.

**3-D Locomotion Task** For the locomotion task, we created three variations of a 4-legged robot such as to mimic the effect of partial damage to one of its legs (Fig. 2). The choice of these morphologies is intended to create a task that would be difficult to master for a neural network that is not able to adapt. During training, both the static-weights and the Hebbian plastic networks follow the same set-up: at each training step the policy is optimised following the ES algorithm described in Section 3.1 where the fitness function consists of the average distance walked of two morphologies, the standard one and the one with damage on the right front leg. The third morphology (damaged on left front leg) is left out of training loop in order to subsequently evaluate the generalisation of the networks.

| Quadruped Damage | Seen / Unseen during training | Learning Rule | Distance travelled | Solved |
|---|---|---|---|---|
| No Damage | Seen | Hebbian | $1051 \pm 113$ | True |
| No Damage | Seen | static weights | $1604 \pm 171$ | True |
| Right front leg | Seen | Hebbian | $1019 \pm 116$ | True |
| Right front leg | Seen | static weights | $1431 \pm 54$ | True |
| Left front leg | Unseen | Hebbian | $452 \pm 95$ | True |
| Left front leg | Unseen | static weights | $68 \pm 56$ | False |

Table 1: Average distance travelled by the highest-performing quadrupeds evolved with both local rules (Hebbian) and static weights, across 100 rollouts. While the Hebbian learning approach finds a solution for the seen and unseen morphologies (defined as moving away from the initial start position at least 100 units of length), the static-weights agent can only develop locomotion for the two morphologies that were present during training.

For the quadruped, we define solving the task as monotonically moving away from its initial position at least 100 units of length along a fixed axis. Out of the five evolutionary runs, both the Hebbian network and the static-network found solutions for the seen morphologies in all runs. On the other hand, the static-weights network was incapable of finding a single solution that would solve the unseen damaged morphology while the Hebbian network did manage to find solutions for the damaged unseen morphology. However, the performances of the Hebbian networks evaluated on the unseen morphology have a high variance. Understanding why some Hebbian solution generalise and other do not paves the way for further research; we hypothesize that in order to obtain a solution capable of generalizing robustly the agent would need to be trained on a diverse set of morphologies with randomized damages. To test how well the evolved solutions generalize, we compare the distance walked averaged over 100 rollouts for the Hebbian and the static-weights networks. We report the highest-performing solutions on each of the morphologies from a single evolutionary run (Table 1).

Since the static-weights network can not adapt to the environment, it solves efficiently the morphologies that has seen during training but fails at the unseen one. On the other hand, the Hebbian network is capable of adapting to the new morphologies leading to an efficient self-organization of network's synaptic weights (Fig. 1). Furthermore, we found that the initial random weights of the network can even be sampled from other distributions than the one used during the discovery of the Hebbian coefficients, such as $\mathcal{N}(0, 0.1)$, and the agent still reaches a comparable performance.

Interestingly, even without the presence of any reward feedback during its lifetime, the Hebbian-based network is able to find well-performing weights for each of the three morphologies. The incoming activation patterns alone are enough for the network to adapt without explicitly knowing which is the morphology currently being simulated. However, for the morphologies that the static-weight network did solve, it reached a higher reward than the Hebbian-based approach. Several reasons may explain this, including the need of extra time to learn or the lager size of the parameters space, which could require longer training times to find even more efficient plasticity rules.

In order to determine the minimum number of timesteps the weights need to converge from random to optimal during an agent's lifetime, we investigated freezing the Hebbian update mechanism of the weights after a different number of timesteps and examining the resulting episode's cumulative reward. We observe that the weights only need between 30 and 80 timesteps (i.e. Hebbian updates), to converge to a set of optimal values (Fig. 3, left). Furthermore, we tested the resilience of the network to external perturbations by saturating all its outputs to 1.0 for 100 timesteps, effectively freezing the agent in place. Fig. 3, right shows that the evolved Hebbian rules allow the network to recover to optimal weights within a few timesteps. Furthermore, the Hebbian network is able to recover from a partial loss of its connections, which we simulate by zeroing out a subset of the synaptic weights during one timestep (Fig. 4, left). We observe a brief disruption in the behavior of the agent, however, the network is able to reconverge towards an optimal solution in a few timesteps (Fig. 4, upper-right).

In order to get a better insight into the effect of the discovered plasticity rules and the development of the weight patterns during the Hebbian learning, we performed a dimensionality reduction through principal component analysis (PCA) which projects the high-dimensional space where the network weights live to a 3-dimensional representation at each timestep such that most of the variance is best explained by this lower dimensional representation (Fig. 5). For the car environment the

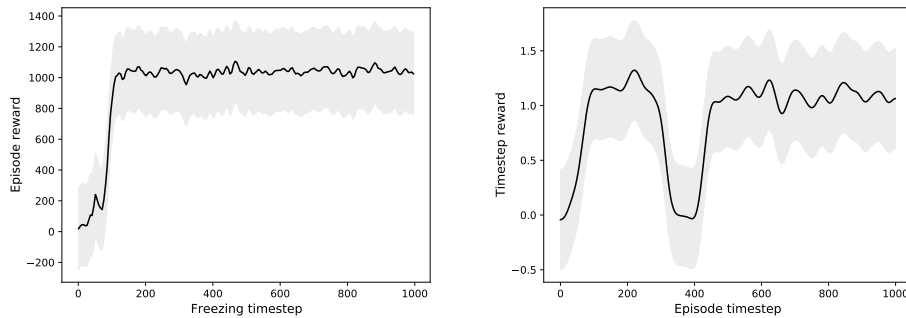

Figure 3: *Learning efficiency and robustness to actuator perturbations.* **Left**: The cumulative reward for the quadruped whose weights are frozen at different timesteps. The Hebbian network only needs in the order of 30–80 timesteps to converge to high-performing weights. **Right**: The performance of a quadruped whose actuators are frozen during 100 timesteps (from t=300 to t=400). The robot is able to quickly recover from this perturbation in around 50 timesteps.

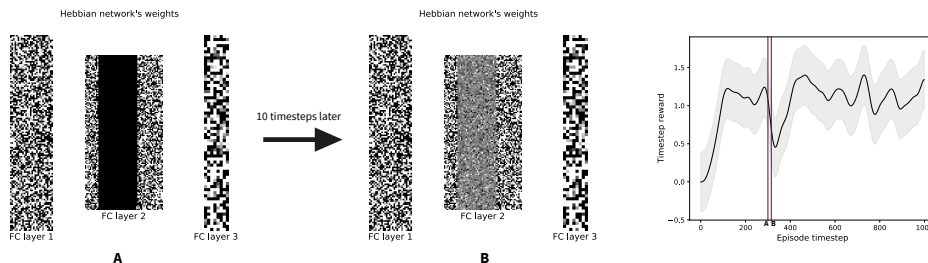

Figure 4: *Resilience to weights perturbations.* **A**: Visualisation of the network's weights at the timestep when a third of its weights are zeroed out, shown as a black band. **B**: Visualisation of the network's weights 10 timesteps after the zeroing; the network's weights recovered from the perturbation. **Right**: Performance of the quadruped when we zero out a subset of the synaptic weights quickly recovers after an initial drop. The purple line indicates the timestep of the weight zeroing.

weights span ubiquitously across the three main components of the reduced PCA space, this contrasts with the dynamics of a network in which we set the Hebbian coefficient (Eq.1) to random values; here the weight trajectory lacks any structure and oscillates around zero. In the case of the three quadruped morphologies, the trajectories of the Hebbian network follow a 3-dimensional curve, with an oscillatory signature; with random Hebbian coefficients the network does not give rise to any apparent structure in its weights trajectory.

## 5  Discussion and Future Work

In this work we introduced a novel approach that allows agents with random weights to adapt quickly to a task. It is interesting to note that lifetime adaptation happens without any explicitly provided reward signal, and is only based on the evolved Hebbian local learning rules. In contrast to typical static network approaches, in which the weights of the network do not change during the lifetime of the agent, the weights in the Hebbian-based networks self-organize and converge to an attractor in weight space during their lifetime.

The ability to adapt weights quickly is shown to be important for tasks such as adapting to damaged robot morphologies, which could be useful for tasks such as continual learning [60]. The ability to converge to high-performing weights from initially random weights is surprisingly robust and the best networks manage to do this for each of the 100 rollouts in the CarRacing domain. That the Hebbian networks are more general but performance for a particular task/robot morphology can be less is maybe not surprising: learning generally takes time but can result in greater generalisation [61].

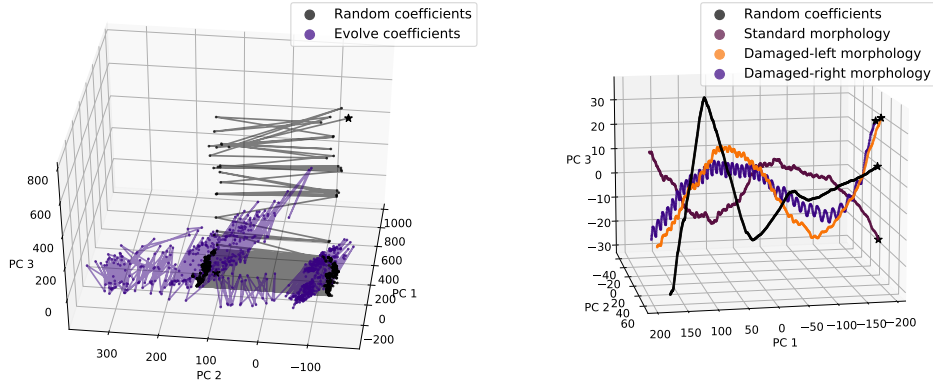

Figure 5: *Discovered Weight Attractors*. Low dimensional representations of the weights dynamics (each dot represents a timestep, first timestep indicated with a star marker). The plotted trajectory represents the evolution of the first 3 principal components (PCA) of the synaptic weights controlled by the Hebbian plasticity mechanism with the evolved coefficients over 1,000 timesteps. Left: Pixel-based CarRacing-v0 agent. Right: The three quadruped agent morphologies: Bullet's AntBulletEnv-v0, the two damaged morphologies [2].

Interestingly, randomly initialised networks have recently shown particularly interesting properties in different domains [16–18]. We add to this recent trend by demonstrating that random weights are all you need to adapt quickly to some complex RL domains, given that they are paired with expressive neural plasticity mechanisms.

An interesting future work direction is to extend the approach with neuromodulated plasticity, which has shown to improve the performance of evolving plastic neural networks [62] and plastic network trained through backpropagation [63]. Among other properties, neuromodulation allows certain neurons to modulate the level of plasticity of the connections in the neural network. Additionally, a complex system of neuromodulation seems critical in animal brains for more elaborated forms of learning [64]. Such an ability could be particularly important when giving the network an additional reward signal as input for goal-based adaptation. The approach presented here opens up other interesting research areas such as also evolving the agents neural architecture [65] or encoding the learning rules through a more indirect genotype-to-phenotype mapping [66, 38].

In the neuroscience community, the question of which parts of animal behaviors are already innate and which parts are acquired through learning is hotly debated [38]. Interestingly, randomness in the connectivity of these biological networks potentially plays a more important part than previously recognized. For example, random feedback connections could allow biological brains to perform a type of backpropagation [67], and there is recent evidence suggesting that the prefrontal cortex might in effect employ a combination of random connectivity and Hebbian learning [19]. To the best of our knowledge, this is the first time the combination of random networks and Hebbian learning has been applied to a complex reinforcement learning problem, which we hope could inspire further cross-pollination of ideas between neuroscience and machine learning in the future [20].

In contrast to current reinforcement learning algorithms that try to be as general as possible, evolution biased animal nervous system to be able to quickly learn by restricting their learning to what is important for their survival [38]. The results presented in this paper, in which the innate agent's knowledge is the evolved learning rules, take a step in this direction. The presented approach opens up interesting future research direction that suggest to demphasize the role played by the network's weights, and focus more on the learning rules themselves. The results on two complex and different reinforcement learning tasks suggest that such an approach is worth exploring further.

## Acknowledgements

This work was supported by a DFF-Danish ERC-programme grant and an Amazon Research Award.

## Broader Impact

The ethical and future societal consequences of this work are hard to predict but likely similar to other work dealing with more adaptive agents and robots. In particular, by giving robots the ability to still function when injured could make it easier for them being deployed in areas that have both a positive and negative impact on society. In the very long term, robots that can adapt could help in industrial automation or help to care for the elderly. On the other hand, more adaptive robots could also be more easily used for military applications. The approach presented in this paper is far from being deployed in these areas, but it its important to discuss its potential long-term consequences early on.

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
