[Supplementary Material]

# 6    Appendix

## 6.1    Network Weight Visualizations

Fig. 6 shows an example of how we visualize the weights of the network for a particular timestep. Each pixel represents the weight value $w_{ij}$ of each synaptic connection. We represent the weights of each of the three fully connected layers *FC layer 1, FC layer 2, FC layer 3* separately: the quadruped's network has an input space of dimension 28 and three fully connected layers with $[128, 64, 8]$ neurons respectively, hence the rectangle above *FC layer 1* has an horizontal dimension of 28 and a vertical one of 128, the 2nd layer *FC layer 2* has an horizontal dimension of 64 and a vertical one of 128 while the last layer's *FC layer 3* dimension is 64 vertical and 8 horizontally, which corresponds to the dimension of the action space. Darker pixels indicate negative values while white pixels are positive values. In the case of the CarRacing environment the weights are normalised to the interval [-1,+1], while the quadruped agents have unbounded weights.

Figure 6: *Network Weights Visualizations.* Visualisation of a random initial state of the network's weights. Each column represents the weights of each of the three layers while each pixel represents the value of a weight between two neurons.

## 6.2    Training efficiency

We show the training over generations for both approaches and both domains in Fig. 7. Even though the Hebbian method has to optimize a significant larger number of parameters, training performance increases similarly fast for both approaches.

(a) Training curves for the Car environment

(b) Training curves for the quadrupeds

Figure 7: **Left:** Training curve for the car environment. **Right:** Training curve of the quadrupeds for both the static network and the Hebbian one. Curves are averaged over three evolutionary runs.

## 6.3 Hebbian rules

We analyze the different flavours of Hebbian rules derived from Eq. 2 in the car racing environment. For this experiment, we do not evolve the parameters of the convolutional layers and instead they are randomly fixed at initialisation; we solely evolve the Hebbian coefficients controlling the feed forward layers. From the simplest one where all but the $A$ coefficients are zero, to its most general form where all the four $A, B, C, D$ coefficient and the intra-life learning rate $\eta$ are present (Fig. 8):

$$\Delta w_{ij} = \eta_w \cdot (A_w o_i o_j + B_w o_i + C_w o_j + D_w), \qquad (2)$$

The static, and all the generalised Hebbian models can solve pixel-based task, only the Hebbian version with a single unique coefficient per synapse $A$ is incapable of solving the task. The slower convergence of the Hebbian models with more coefficients can be explained by the fact that larger parameter spaces need more generations to be explored by the ES algorithm.

Figure 8: *Hebbian rules ablations.* Training curves for the car racing agent with five different Hebbian rule variations. Curves are averaged over three evolutionary runs.

We also show the distribution of coefficients of the most general ABCD+$\eta$ version (Fig. 10), which shows a normal distribution. We hypothesise that this distribution is potentially necessary to allow the self-organization of weights to not grow to extreme values. Analysing the resulting weight distributions and evolved rules opens up many interesting future research directions.

## 6.4 Evolving initial weights and learning rules

We experimented with evolving –alongside the Hebbian coefficients– the initial weights of the network rather than randomly initializing them at each episode. We do this by sampling normal noise twice (Algorithm 2, Step 5 from [44]) and computing the fitness of the resulting solution pairs (Hebbian coefficients, initial weights). Surprisingly, this does not increase the training efficiency of the agents (Fig. 9). Furthermore, we find that runs for the CarRacing environment where we co-evolve the initial conditions are more likely to stall on local optima: 2 out of 3 runs found a network with good performance (at least 800 reward), while the third run stalled on low performance (a reward of less than 100). This finding may be explained by the extra difficulty that co-evolution introduces in the ES algorithm as well as the extra *lottery-ticket* initialisation of both the initial weights and the Hebbian coefficients [68]. However, other possible implementations of this system may yield better results and evolving both the connections' Hebbian coefficients and learning rules has shown promise in smaller networks [45, 13, 66].

Figure 9: Training curves of the Hebbian networks for the quadruped environments. We show that initializing the network with random weights at each episode and co-evolving the initial weights lead to similar results. Curves are averaged over three evolutionary runs.

Figure 10: Distribution of Hebbian coefficients for the Hebbian network solutions of the quadrupeds (left) and the racing car (right).