[Reviews · NeurIPS 2020]

Review 1

Summary and Contributions: The authors use an evolutionary meta-learning approach to determine the parameters of a Hebbian plasticity rule that allows networks to adapt from random weights to perform tasks, as opposed to learning weights themselves. The tasks used are RL tasks: a car racing and quadruped task. Parameters of a Hebbian-like learning rule, which can be different for each synapse, are optimized. The method performs competitively on the tasks and there is an argument for increased robustness to perturbations.

Strengths: This is an interesting avenue for biologically plausible learning, as the rules being studied make no requirements on biologically implausible weight transport. The results are competitive with some deep RL approaches. The fact that per-parameter plasticity rules are optimized means the work is an extension beyond other approaches that focus on uniform plasticity rules across neurons. Although this raises questions about biological plausibility and overparameterization, it means the models are more flexible and powerful than prior related work.

Weaknesses: The fact that every neuron's plasticity parameter can be learned makes it difficult to interpret what is being learned. Are the weights effectively learning to relax to the same steady state from random initial conditions (in which case the plasticity rules are essentially encoding weights)? The illustrated weight attractors do not provide much insight. The results for average distance traveled are not particularly convincing because the static weight networks outperform the Hebbian weights so drastically for two out of three situations. The requirement that networks evolve from random initial weights may be limiting performance, and from a biological standpoint completely random weights without any structure is probably not the appropriate starting point (evolution and development may optimize this initial condition). It would be useful to examine what happens when both initial condition and plasticity rule are optimized. Does this retain the high performance of the static weight networks while still permitting flexibility? This would also more effectively isolate benefits that are due solely to plastic Hebbian weights from those that are due to something like an initial condition.

Correctness: The simulations appear to be correctly implemented.

Clarity: The paper is clearly written.

Relation to Prior Work: The manuscript has a good review of prior meta-learning literature.

Reproducibility: Yes

Additional Feedback: After discussion and author response, I still feel the paper is marginally above the acceptance threshold.


Review 2

Summary and Contributions: The authors used an evolutionary algorithm to meta-learn Hebbian-like plasticity rules of neural networks in two reinforcement learning environments. The authors demonstrated that this approach (1) works in these environments, (2) provides networks that could adapt better to variations in robot morphologies. Understanding biological plasticity rules and how these rules enable animals to be so adaptive is an important topic. The authors provide an interesting approach to this challenge.

Strengths: The authors tested their evolutionary approach on interesting RL problems, and obtained sensible results. Evolutionary approach is not a commonly used method in machine learning (compared to end-to-end gradient-based deep RL), and the field would benefit from understanding better the strength and weakness of this approach to modern problems. The authors evolved the synaptic plasticity parameters of all plastic connections in the network, making the optimization problem very high dimensional. It’s encouraging to see that the evolutionary algorithm still works reasonably well in this context.

Weaknesses: It would be quite interesting if the authors could expand more their analysis of the evolved plasticity rules (not just the weights as in Fig 5). What kind of plasticity rules were discovered by the evolution algorithm? The authors could perhaps make space for such analysis by reducing the length of the Discussion section. Although quantitative performance is not and should not be the main focus of this work, it would be nice if the authors provide a more systematic summary of the performance of other approaches on the two environments studied. The authors did some of this in the text (like line 209), but it would be nice to show this in a table. I don’t feel too strongly about this point though, because emphasizing the quantitative performance may distract readers from the bigger message.

Correctness: The equation between line 149 and 150 has a mathematical error. The RHS of the $h_t+1$ update should use the instantiated noise, not a new random variable (N(0, 1)). Please see Algorithm 1 in Salimans et al. 2017 for the mathematically correct formulation. I checked the code provided by the authors, and the implementation appears to be correct, so this error shouldn’t impact the authors’ results.

Clarity: The paper is overall clear and understandable. Typo on line 207: state-of-the-art Figure 5 left, the “first time step indicated with a star marker” is difficult to see.

Relation to Prior Work: I think the paper is well-referenced, with a healthy balance of modern and classical papers. Line 40, some references are warranted for the statement “While recent work in this area has focused on determining both the weights of the network and the plasticity parameters”. In the related work section, the authors should cite and discuss some recent papers from Sohl-Dickstein, for example Metz, Maheswaranathan, Cheung, Sohl-Dickstein 2018.

Reproducibility: Yes

Additional Feedback: ***Update after rebuttal*** I was interested in seeing more analyses about the learned plasticity rules. The authors analyzed the correlation between the plasticity rule parameters (A,B,C,D) (rebuttal fig 1d). I don't think this analysis is informative. At the same time, I don't think I have new concerns after reading the reviews of other reviewers (it seems like there is no fundamental disagreement among reviewers, just preferences). Overall, my score stays the same (7, accept).


Review 3

Summary and Contributions: Parameters of Hebbian synaptic plasticity rules are optimized by Evolution Strategies for two tasks. In particular, it is shown that this method enables fast recovery from damage for a simulated 3D quadrupedal robot.

Strengths: There exists substantial evidence from neuroscience that different types of synaptic connections in the brain tend to be controlled by different plasticity rules, rather than by a single one. Hence it is quite convincing to assume that evolution has optimized them to support fast learning in situations that are essential for survival. The authors use the power of advanced ES strategies to emulate evolutionary optimization, and achieve nice performance for the two tasks that are considered.

Weaknesses: I find it difficult to relate the Hebbian plasticity rules that are considered in this paper to rules for synaptic plasticity in the brain that have been found in neuroscience. Synaptic plasticity in the brain appears to rely often on a multitide of gating signals, and on the relative timing of pre- and postsynaptic activity. Also the recent history of a synapse appears to play a role. Hence I find it difficult to convince myself that this paper provide new insight into synaptic plasticity or the organization of learning in the brain. The overall conceptual approach follows the one proposed by ref. 30. Hence the originality of this paper lies more in the use of more advanced optimization tools, which enable in particular scaling up to larger models.

Correctness: Yes, as far as I can see.

Clarity: Yes.

Relation to Prior Work: yes

Reproducibility: Yes

Additional Feedback:


Review 4

Summary and Contributions: The paper presents a approach to evolve/meta-learn hebbian plasticity parameters of policy networks instead of the weights themselves. It is shown that with the evolved plasticity rules the policy networks can perform well even when starting from random initial conditions.

Strengths: - The paper follows an interesting idea. Studying life-time adaptive networks is important - The approach is sufficiently novel from what I can tell - Illustrations of some of the inner workings - The paper is easy to understand

Weaknesses: - I think the claims about improved generalization with the hebbian learning cannot be made, because the system was optimized for different morphologies. - The adaptation to unseen morphological changes would have been very convincing. - Also the argument that now we can use random networks and the plasticity would be innate is also not very strong because every neuron has its own specific rule. The number of parameters of the system is thus 5 times larger than the ones of the static network. I suggest to have an ablation where the hebbian parameters are shared among the neurons in one layer. - It is very strange that all realizations would be bad at the same morphology for the ant. see below.

Correctness: EQ after 149 (should have a number): The N(0,1) term here is misleading and I think it is wrong. What is meant is the same deviation sample that was used in h_t+sigma N(0,1). I recommend to use \xi_t = N(0,1) and then you can use the same noise realization in both equations. I am questioning the results on the Ant for symmetry reasons: To me these two changed morphologies are essentially equal up to a rotational symmetry and the fitness function is also invariant to this symmetry. So I expect that each run of the static network would be bad at one of the mophologies and not all the time at the same.

Clarity: The paper is well understandable. line 142 and following: I think h is the set {A,B,C,D}, should be explicitly stated I was not sure whether the Ant was trained on all morphologies. I found it in the text, but maybe this could be more prominent. Some sentences suggest that there is an "adapation" to the other mophologies, but the policy always "adapts" to the current situation, but all of them are seen during training.

Relation to Prior Work: The relation to prior work is well done. A more general hebbian learning rules are described in this paper: Zappacosta S, Mannella F, Mirolli M, Baldassarre G (2018) General differential Hebbian learning: Capturing temporal relations between events in neural networks and the brain. PLOS Computational Biology 14(8): e1006227 which would give you more than 4 parameters to tune. Along the lines of differential Hebbian Learning this paper Der, R., Martius, G. Novel plasticity rule can explain the development of sensorimotor intelligence. Proceedings of the National Academy of Sciences, 112(45):E6224-E6232, 2015 is also relevant, as it shows how a learning rule can generate a variety of coordinated behavior even without optimizing it for the morphology or specializing it for each neuron.

Reproducibility: Yes

Additional Feedback: The code is provided, however exact call-commands with the parameters used was not provided. Fig 4: it looks more like 100-200 steps to converge The color codes in the figures are hard to read/distinghuish *** UPDATE AFTER REBUTTAL *** thanks for the clarifications. I suggest to include the results that you obtained by leaving out morphologies at training time in the final version.

[Author Response · NeurIPS 2020]

**Reviewer 1** asked if the weights are effectively learning to relax to the same steady state from random initial conditions. We performed an analysis to be added to the appendix (Fig. 1a,b), which shows that for two different runs the weights start being perpendicular –since randomly initialised– and thereafter the $n$-dimensional angle between the weights is not zero. Thus the weights are not effectively relaxing to a fixed set of values but rather change based on the environmental input. Additionally, the absolute weights of a particular network keep changing as well (Fig. 1c)

| (a) Quadruped: weight angle | (b) Car: weight angle | (c) Car: magnitude change | (d) Coefficient Correlations |

Figure 1

Following **Reviewer 1**'s suggestion of having both the initial condition and plasticity rules optimized, we performed 6 evolutionary runs in the car-racing environment; 3 of them converged to high-performance solutions ($879 \pm 69$), and 3 of them did not ($13 \pm 2$). This suggest that evolving both at the same time could be more challenging (all random+Hebbian runs always found a high-performing solution). We are currently running the same experiments for the ant, but they are unfortunately not finished yet.

**Reviewer 2** noted that "it would be quite interesting if the authors could expand more their analysis of the evolved plasticity rules." We performed a correlation analysis of the evolved Hebbian coefficients and found no correlation between them –neither Pearson's $r$ nor Spearman's $\varrho$–, suggesting that the coefficients are independent with no obvious internal structure in the learning rules (Fig. 1d). We are currently investigating if frequent pattern mining methods (e.g. apriori algorithm) could reveal additional insights.

**Reviewer 3** said to "find it difficult to relate the Hebbian plasticity rules that are considered in this paper to rules for synaptic plasticity in the brain that have been found in neuroscience" arguing that "synaptic plasticity in the brain appears to rely often on a multitude of gating signals, and on the relative timing of pre- and postsynaptic activity." The most studied plasticity mechanism in neuroscience is spike-timing-dependent plasticity (STDP). However, STDP isn't the only plasticity mechanism that has been observed in the brain.

NNs with continuous outputs are usually interpreted as an abstraction of spiking neural networks in which the continuous output of each neuron represents a *spike-rate coding* average (instead of spike-timing coding) of a neuron over a long time window or, equivalent, of a subset of spiking neurons over a short time window (in this scenario, the relative timing of the pre and post-synaptic activity doesn't play a central role anymore [1]. Spike-rate-dependent plasticity (SRDP) is well documented phenomena in biological brains [2][3]. That being said, our goal is not to provide a detailed model of plasticity mechanisms in biological brains, but rather to demonstrate that evolved local rules can show adaptability and yield competitive results. Additionally, in contrast to earlier work that restricted learning rules to only four different hand-designed types (ref [30]), we evolve arbitrary synapse-specific Hebbian rules.

**Reviewer 4** objected to the claim that the Hebbian approach led to greater generalization due to the fact the system was optimized for all the morphologies. Following this fair critique, we launched a new set of experiments where one of the damaged morpgologies was left out during training. While the static network only managed to solve the morphologies it had seen during the training phase, the Hebbian network –to our surprise– managed to solve all three of them in each of 3 different evolutionary runs. On the unseen morphology the Hebbian approach reached a performance of $471 \pm 87$ compared to the static-network performance of $31 \pm 46$. We believe that this adaptation to the unseen morphological changes provides evidence of greater generalisation for the Hebbian network in relation to the static-weights one.

We ran an ablation where the Hebbian parameters are shared among neurons in one layer for the car environment: all 3 evolutionary runs resulted in poor performance (max reward $13 \pm 2$ as opposed to $870 \pm 13$ with individual learning rules). This suggests that a higher number of parameters is important for the agents to adapt. **Reviewer 4** argued that due to the rotational symmetry of the ant, the under-performing runs shouldn't always be the same morphology: the reported results are averages across 100 rollouts of the best solution, which had visibly evolved to favour one of the damaged morphologies; in other evolutionary trains the favoured morphology changed, which we will clarify in the paper. We finally thank the reviewers for their careful read and will subsequently correct the equation after line 149.

[1]Brette, 2015. *Philosophy of the Spike: Rate-Based vs. Spike-Based Theories of the Brain.* Frontiers in Systems Neuroscience

[2]Sjöström et al., 2001. *Rate, timing, and cooperativity jointly determine cortical synaptic plasticity.* Neuron

[3]Prescott et al., 2008. *Spike-Rate Coding and Spike-Time Coding.* Journal of Neuroscience


[Meta-Review · NeurIPS 2020]

The paper proposes the use of evolution to find the params of a (parametrized) Hebbian plasticity learning rule and optimize the network to be able to adapt from random weights to perform tasks rather than learning weights themselves. They experiment with car racing and robot simulation tasks. The paper is well written, and the idea is interesting with motivations from neuroscience. Reviewers generally find the work encouraging, and suggested improvements / future work such as looking at generalization abilities. R3 also made a good point that while the work is motivated from neuroscience, it is “difficult to relate the Hebbian plasticity rules that are considered in this paper to rules for synaptic plasticity in the brain that have been found in neuroscience. Synaptic plasticity in the brain appears to rely often on a multitude of gating signals, and on the relative timing of pre- and postsynaptic activity. Also the recent history of a synapse appears to play a role. Hence I find it difficult to convince myself that this paper provide new insight into synaptic plasticity or the organization of learning in the brain.” A study of why the current proposed method works will be required to understand whether it overly relies on the optimization method used, and perhaps shed some new light on the role of synaptic plasticity in meta-learning.